# Microstructure and Mechanical Properties of HfC-SiC Ceramics Influenced by WC Addition

**DOI:** 10.3390/ma16093337

**Published:** 2023-04-24

**Authors:** Yue Cheng, Huaguo Tang, Guangkai Fang, Yuan Yu, Lujie Wang, Yanfei Zhang, Zhuhui Qiao

**Affiliations:** 1School of Mechanical Engineering, Qilu University of Technology, Shandong Academy of Science, Jinan 250353, China; 2Shangdong Laboratory of Yantai Advanced Materials and Green Manufacture, Yantai 264006, China; 3Yantai Zhongke Research Institute of Advanced Materials and Green Chemical Engineering, Yantai 264006, China; 4State Key Laboratory of Solid Lubrication, Lanzhou Institute of Chemical Physics, Chinese Academy of Sciences, Lanzhou 730000, China

**Keywords:** HfC-SiC-WC ceramics, solid solution, mechanical properties, SPS

## Abstract

The development of HfC-SiC has been challenging due to difficulties in achieving sintering and satisfactory mechanical properties. However, this study aims to overcome these limitations by incorporating WC as an additive. SPS was employed to process HfC-SiC and HfC-SiC doped with 5 vol.% WC. The resulting samples were then evaluated for their oxygen content, relative density, Vickers hardness, bending strength, indentation fracture toughness, and microstructure. The Vickers hardness (20.50 ± 0.20 GPa), flexural strength (600.19 ± 84.00 MPa), and indentation fracture toughness (5.76 ± 0.54 MPa·m^1/2^) of HfC-30 vol.% SiC-5 vol.% WC ceramics are higher than HfC-30 vol.% SiC ceramics. Doping 5 vol.% WC in HfC-30 vol.% SiC not only reduces the oxygen content of samples but also produces the (Hf,W)C solid solution and refines the microstructures, which are the main reasons for the higher mechanical properties of HfC-30 vol.% SiC-5 vol.% WC ceramics. In summary, this study successfully addresses the challenges associated with HfC-SiC by incorporating WC as an additive, leading to improved mechanical properties and microstructures.

## 1. Introduction

Ultra-high temperature ceramics are a class of materials with outstanding mechanical and thermochemical properties and potential for various applications, including leading edges of aerodynamic propeller or wing structures, the nose cone of hypersonic reentry vehicles, rocket nozzles, and thermal protection systems, which need to operate in harsh environments [1,2,3,4,5,6]. HfC is a kind of ultra-high temperature ceramic, with melting higher than 2000 °C, high hardness (HfC: 22.1 GPa [7]), excellent elastic modulus at room temperature (HfC: 461 GPa [8]), and it also has splendid resistance to chemical corrosion and thermal shock [9,10,11]. Nevertheless, despite its suitable properties, monolithic HfC is difficult to sinter due to its low self-diffusion coefficient and strong covalent bonds. These characteristics signify that it needs a high sintering temperature (usually above 2000 °C) to increase sintering driving force, which leads to the rapid growth of grains, coarse structure, and poor mechanical properties [7,12].

To solve the above problems, some scholars [7,13,14,15,16,17,18,19,20,21,22,23,24,25,26,27] carried out work with regard to improving sintering density and grain refinement using HfC with particle size distribution between 0.2 and 1.5 μm as the starting powder, and adopted the pressure-less sintering at 1950 °C, keeping it for 1 h; however, the relative density of HfC ceramics was as low as 70% [13]. Even after raising the sintering temperature to 2760 °C and keeping it for 2 h, the relative density of sintered HfC ceramics was only 90% [14]. To improve the relative density, HfC ceramics were usually sintered at high temperatures with external pressure [12,15]. Hot pressing (HP) sintering obtained 98% HfC ceramics with a grain size distribution between 40 and 60 μm at 2000–2500 °C for 2 h [12]. Using SPS technology, HfC ceramics with the relative density of 98% and average grain size of 19 μm were obtained over 2400 °C at 65 MPa, and a sintering time of 3–15 min [7]. The results show that high temperature and high pressure sintering or SPS can supply the sufficient sintering driving force for the densification, and also lead to the rapid growth of HfC. To prevent the crystalline grain growth during sintering, we often add a second phase into the matrix, and SiC is usually used as a secondary phase for ceramics to adjust the microstructure of ceramics and enhance their performance because of its stable chemical properties, high hardness, and excellent oxidation resistance [4,16,17,18,19]. Moreover, in SiC powder with finer particle size, the refinement effect is stronger, densification promotion efficiency is higher, mechanical properties and the performance are better than coarse SiC powder [16,20]. In the previous study, ZrB_2_-SiC ceramics were prepared with ZrB_2_ as the raw powder and WC as the sintering aid to improve the three-point bending strength at high temperatures [21,22]. The three-point bending strength of ZrB_2_-SiC ceramics at 1600 °C was 460 MPa, but the three-point bending strength of the sample doped with WC was increased to 675 MPa. The results show that C, B_4_C, WC, and other refractory ceramic additives can improve the densification behavior of ZrB_2_ ceramics with the method of eliminating the surface oxides of ZrB_2_ and activating the raw powder [23,24,25,26,27]. The HfB_2_-SiC ceramics exhibited creep deformation and the measured bending strength of 389 ± 82 MPa at 1600 °C, while the samples with WC doped exhibited higher bending strength. The elimination of oxide impurities at grain boundaries and the formation of solid solution were considered to be the main reasons for the performance improvement [22].

The addition of suitable additives to ceramic materials is an effective way to improve their mechanical properties. The above studies had established that WC can significantly enhance the mechanical properties of ceramic materials. However, the application of WC as an additive to improve the microstructure and mechanical properties of HfC-SiC ceramics has not been extensively explored. Therefore, a novel study was conducted to investigate the effect of the addition of 5% WC on the microstructure and mechanical properties of HfC-SiC ceramics. This study aims to contribute to the development of high-performance ceramic materials with improved mechanical properties for a wide range of industrial and technological applications.

## 2. Experimental Process

### 2.1. Raw Materials and Powder Processing

Both HfC and SiC powders are produced by Changsha Tianjiu Metal Material Co, LTD (Changsha, China). Table 1 lists the purity, major impurities, and average particle size of all raw materials.

In order to study the effect of WC additive on the microstructure and properties of HfC-SiC ceramics, two different components of HfC-30 vol.% SiC (HS) and HfC-30 vol.% SiC-5 vol.% WC (HSW) were prepared. The raw materials were milled for 10 h in planetary ball mill (Nanda Instrument, QM-3SP4) with Si_3_N_4_ balls and alcohol as a medium to obtain the mixed power. Then the pellets were dried in a vacuum drying oven at 80 °C for 10 h. After drying, powders were prepared through a sieve of 200 mesh. The graphical experimental procedure for the sintered sample is shown in Figure 1.

### 2.2. Sintering and Polishing

#### 2.2.1. Sintering of the Samples

The samples were sintered in a plasma spark sintering furnace. A pre-pressure of 14.15 MPa was applied before sintering to compact the raw material powder. At the same time as the pre-pressure loading, the vacuum degree in the furnace was pumped to 15 Pa to prevent oxygen contamination during the sintering process. During sintering, the temperature was increased from room temperature to 700 °C within 4 min, and the pressure was increased from 14.15 to 32 MPa. Next, increasing the temperature from 700 to 1400 °C, the pressure was kept at 32 MPa during heating, and the heating rate was kept at 200 °C/min. When the temperature increased from 1400 to 2100 °C, the pressure increased to 50 MPa and the heating rate was kept at 200 °C/min. Rapid heating can prevent grain growth to some extent. The sample was further densified by holding it at 2100 °C for 20 min, and then the temperature was raised to 2300 °C within 2 min and kept for 20 min for the solid solution to occur adequately [8,28].

#### 2.2.2. Polishing and Processing of the Samples

After sintering and cooling the furnace to room temperature, the sample was ground and polished to 0.5 μm finish by polishing the graphite surface with a sand paper. The polished samples were cut into test rods by diamond wire cutters, and the surfaces of the test rods were polished using a diamond polishing paste. The size of the test rod was 3 mm × 4 mm × 25 mm. Using anhydrous ethanol as a medium, the test rod was cleaned ultrasonically. After ultrasonic cleaning, the rods were dried for characterization. Five rods were tested in each experiment.

### 2.3. Characterization of Samples and Performance Testing

#### 2.3.1. Characterization of Samples

The relative densities of the specimens were determined using the Archimedean drainage method. The theoretical densities were calculated from different proportions of the components. The oxygen content of the specimens was determined by oxygen nitrogen hydrogen analyzer (ONH836, RORIBA, Kyoto, Japan). X-ray diffraction (XRD, D8 Advance, Bruker, Beijing, China) was used to analyze the diffraction peaks of the mixed powder and sintered sample. Energy-dispersive spectroscopy matching (EDS, Zeiss Ultra +, Carl Zeiss, Jena, Germany) with SEM was used to analyze the phase composition and related element distribution of the sintered sample. The microstructure of the polished surface and the grains fracture mode of the cross-section was observed by a scanning electron microscope (SEM, Zeiss Ultra +, Carl Zeiss, Jena, Germany).

#### 2.3.2. Performance Testing

Vickers hardness (HV) was determined by Vickers indentation (HV-1000Micro Vickers Hardness Tester, Love Measurement Technology Co, Ltd, Hong Kong, China) under the load of 1 kg and dwell time of 15 s. Flexural strength tests were carried out on the Instron-5969 universal testing machine, using 20 mm as the span on the 3 mm × 4 mm × 25 mm sample strip, and the crosshead drop rate was 0.5 mm/min. The bending strength at room temperature was calculated using Equation (1).
(1)$$\sigma =\frac{3PL}{2bh^{2}}$$
where *σ* is the bending strength (MPa), *P* is the load at fracture (N), *L* is the sample span (mm), *b* is the sample width (mm), and *h* is the sample height (mm). Indentation fracture toughness (*K_I__C_*) was determined by the indentation method with a load of 5 kg and a dwell time of 10 s in Vickers indentation. The samples were pressed out of cracks and indentations, and the indentation fracture toughness of the material was calculated by Equation (2); the elastic modulus required for calculation is obtained from Equation (3).
(2)$$K_{Ic}=0.016{\left(\frac{E}{HV}\right)}^{\frac{1}{2}}(\frac{P}{C^{\frac{3}{2}}})$$
where *K_IC_* is the indentation fracture toughness (MPa·m^1/2^), *E* is the elastic modulus of the sample (GPa), *HV* is the Vickers hardness of the sample (GPa), *P* is the applied load (N), and *C* is the average length of the diagonal crack (mm).
(3)$$E=\frac{3L(P_{2}-P_{1})}{2bh^{2}(g_{1}-g_{2})}$$
where *E* is the elastic modulus (GPa); *L* is the sample span (mm); *P_i_* is the load at two points on straight line segment of the load–strain curve (N), i = 1,2; *b* is the width of the sample (mm); *h* is the height of the sample (mm); *g_i_* is the micro-strain measured at the load, i = 1,2. All mechanical properties values are the average values of five measurements.

## 3. Results and Discussion

### 3.1. Oxygen Content

Oxygen contamination is the main reason for the suppression of UHTC densification in carbon-based UHTCs. The impurities on the surface of the particles hinder the diffusion-driven mass transfer mechanism during the consolidation process and coarsen the grains [29,30,31]. To accelerate the densification of the HS, the oxide impurities must be removed before densification.

The residual oxygen content of HS and HSW was measured by an oxygen nitrogen hydrogen analyzer (ONH836). The measurements show a residual oxygen content of 1.8543 wt.% for HS and 0.8894 wt.% for HSW. It has been shown that for HS, the addition of WC favors the removal of oxide impurities in the sintering process.

The oxide impurities on the surface of HfC and SiC particles are most likely in the form of HfO_2_ and SiO_2_. SiO_2_ impurities can be removed by a reaction between SiC and SiO_2_ at 1700 °C. This process is referred to as the self-cleaning of SiC [32]. In the sintering temperature range, HfO_2_ has very low vapor pressure with a melting point of 2810 °C, which is difficult to remove by vaporization. Some sintering additives need to be added to remove HfO_2_. Previous studies have shown that in pressure-less sintering at high temperature, WC can react with HfO_2_ and remove impurities through reaction (4) [33]:(4)$$\begin{array}{c} HfO_{2}+3WC\to HfC+3W+2CO(g)\\ \left(T>1260{}^{\circ}C,P_{CO}=10Pa,\Delta G<0\right) \end{array}$$

According to thermodynamic calculation, the reaction (4) runs smoothly above 1200 °C at CO(g) partial pressure of 10 Pa. This reaction effectively reduces the HfO_2_ impurities, resulting in a low residual oxygen content of the HSW.

### 3.2. Densification and Densification Behavior

The relative densities of SPS sintering HS and HSW are 95.87 ± 0.25% and 96.92 ± 0.18%. It is shown that the sintering properties of the initial powder are favorable. Figure 2 shows the displacement–temperature–time (DTT) curve during the sintering process of HS and HSW. Figure 2 shows that the HSW displacement curve starts to decline earlier, indicating that the densification starting temperature of HSW is relatively lower than HS [34].

In the DTT diagram, the whole sintering process can be divided into four parts. In the first region, the displacement is positive and manifests itself as an upward motion. The cause of this is the thermal expansion of the graphite molds, gaskets, and powders used in the sintering process when the temperature is raised rapidly. At this stage, the powder starts to shrink. In the second region, the powder continues to shrink. The particles rearrange under pressure and Joule heating, and the expansion of the inter-particle contact region generates more sparks and Joule heating, forming a neck. In the third region, the displacement changes dramatically at short times, which is due to the further densification of the powder at high temperature and pressure. The bulk deformation, the neck growth, the sufficient contact of the particles, and the apparent contraction occur in this region due to the increase in temperature and pressure. In the fourth region, the displacement is slightly increased and the sintering is completed. In this sintering study, neck formation, local deformation, neck growth, and subsequent full particle contact are typical solid-state densification mechanisms, indicating that both HS and HSW are solid-state densification mechanisms [35,36,37]. Figure 3a shows that there is no secondary or residual phase at the interface, indicating that no liquid phase is formed at the interface. This result proves that solid phase diffusion is the mass transfer mechanism in HS sintering process.

### 3.3. Phase Composition and Microstructure of Sintering Ceramics

Figure 4 shows the SEM images of the polished surface and SEM images at high magnitudes for HS and HSW. The results of EDS show that HfC grains in HSW contain the W element (see Figure 3c,d). According to Figure 5a,b and XRD data, the diffraction peak position of HfC in HS is 56.065°, and the lattice parameter is 4.638A, while the HfC diffraction peak position of HSW is 56.790°, the lattice parameter is 4.568 ± 0.014 A. The diffraction peak shifts to a higher position and the lattice parameter changes are due to the formation of (Hf,W)C solid solution and the lattice distortion caused by the addition of WC [33,38]. In addition, lattice distortions can reduce the activation energy of the sinter and simultaneously improve the density. Because the pores are enclosed within the grain, it is not possible to measure the porosity using traditional boiling. Therefore, the porosity has to be deduced from the SEM micrographs. As shown in Figure 3b,d, we can find some tiny pores in both HS and HSW, with HS having a higher porosity than HSW, and the pores in HSW are difficult to identify. Because of the formation of (Hf, W)C solid solution, the densities of HSW could not be precisely determined using the theoretical densities calculated from the densities of the starting components. Moreover, the reason that pores in HSW are difficult to identify is that the actual relative density is slightly higher than that measured by the experiment [39].

The XRD patterns of sintered ceramics show that HfC and SiC are the main phases (see Figure 5a). The experiment result shows that HS is stable at 2300 °C. This is consistent with the phase diagram of HfC-SiC system which shows that HfC and SiC are stable solid phases before 2637 °C [14]. As can be seen from Figure 5a, the HfO_2_ content in the HSW significantly decreases with the addition of WC, again demonstrating the deoxygenation effect of WC. Figure 4b,d show that HS and HSW mainly contain a gray phase, a light gray phase, and a black phase. To determine their composition, the analysis was performed using XRD and EDS separately. The results show that the gray phase and light gray phase in HS are HfC and the black phase is SiC, while the gray phase and light gray phase in HSW is (Hf,W)C solid solution and the black phase is SiC. Through the analysis of the content of each element in Figure 3, the percentage of SiC in Figure 3 is 56.48%, and the percentage of (Hf,W)C is 43.52%. As can be seen from Figure 4b,d and Figure 6a,b, the microstructure of HSW is relatively fine, and the addition of WC refines the microstructure of HSW [33,34].

### 3.4. Mechanical Properties

Figure 7 shows the relative density and mechanical properties of HS and HSW. In Figure 7a, the addition of WC increases the relative density of the sample from (95.88 ± 0.25) to (96.92 ± 0.18)%. As mentioned in the previous studies, the oxide impurities on the particle surface are the most important factors affecting the relative density [29,30,31]. In order to improve the relative density of the sample, it is necessary to remove the oxide impurities from the [30] particle surface as far as possible. However, WC can achieve the purpose of eliminating the surface oxides and activating the raw powder by reacting (4) with HfO_2_. Other researchers have reported the positive effects of the addition of MC (M = Zr, Ti, V...) on the sintering properties of MB_2_ composites, including the reduction of apparent porosity and the promotion of plastic deformation on surface and neck formation, all of these will promote the relative density to some extent [40,41,42].

Similarly, in Figure 7b, the addition of WC also improves the flexural strength of the sample from 486.48 ± 5.05 to 600.20 ± 84.13 MPa. The increase in flexural strength is put down to an increase in relative density and a decrease in grain size. As shown in Figure 6a, there are some SiC clusters in HS ceramics. Large SiC clusters could result in high residual stress and stress concentration, which is harmful to the mechanical properties [21,43,44,45]. It can be observed from Figure 6b that the HSW has a uniform structure and particle distribution, which can achieve better mechanical properties.

In Figure 7c, with the addition of 5 vol.% WC, the Vickers hardness of the sample increases from 19.10 ± 0.21 to 20.51 ± 0.21 GPa. A large number of studies show that the Vickers hardness of ceramics is related to density closely [46,47,48]. Generally speaking, the Vickers hardness of ceramics increases with the increase in density, and HS also follows this law [8,39]. As shown in Figure 3b,d, some tiny pores can be observed in HS and HSW, with HS having a higher porosity than HSW. As the porosity decreases, the Vickers hardness of the HSW increases. In addition, the hardness of WC (about 24 GPa) is higher than that of HfC-30 vol.%SiC (about 20 GPa [14]) and the formation of solid solution, which also leads to a higher hardness of HSW than HS.

In Figure 7d, the Indentation Fracture toughness of HS is 4.88 ± 0.18 MPa·m^1/2^ while that of HSW is 5.76 ± 0.54 MPa·m^1/2^, the indentation fracture toughness of HSW is 17.8% higher than HS because of the addition of WC. To explain the toughening mechanism, the crack propagation paths on polished surfaces and fracture sections of HS and HSW are observed by SEM (Figure 8). Figure 8a shows that the fracture mode of HS is mainly the combination of transgranular and intergranular (shown with narrow arrows), and the crack deflection and transgranular can be observed in the fracture section (see Figure 8c). Figure 8b shows that the HSW fracture mode is mainly intergranular with a small amount of transgranular (shown with thick arrows). In Figure 8d, the crack deflection, crack bridging, and crack branching are clearly observed. Therefore, the mechanism of toughening can be confirmed by crack propagation and fracture patterns. The crack deflection causes the crack propagation to become jagged, thereby increasing the actual crack surface area and extending the crack propagation length. And the crack bridging creates a closing stress at the crack, which creates a tendency to connect the two cracked surfaces and bring the two surfaces together. At the same time, crack branching generates microcracks to disperse the energy at the tip of the main crack, which is consumed in the propagation and formation of new microcracks to achieve macro-toughening. In general, crack deflection, crack bridging, and crack branching all increase the energy dissipation during the fracture process and thus increase the indentation fracture toughness of materials [34,37,49].

## 4. Conclusions

In this study, HfC-30 vol.% SiC containing 5 vol.% WC and HfC-30 vol.% SiC without WC are sintered by SPS at 2300 °C, respectively, solving the problems of achieving sintering and satisfactory mechanical properties in HfC-30 vol.% SiC ceramics. The effect of 5 vol.% WC on the microstructure and mechanical properties of ceramics has been studied. The conclusions are as follows:(1)In terms of microstructure, the sintered HfC-30 vol.% SiC and HfC-30 vol.% SiC-5 vol.% WC are all composed of three phases. The difference is that HfC-30 vol.%SiC-5 vol.%WC generated (Hf,W)C solid solution. At the same time, there are some SiC clusters in HfC-30 vol.% SiC ceramics, while there are no clusters in HfC-30 vol.% SiC-5 vol.% WC ceramics, and the microstructure of HfC-30 vol.% SiC-5 vol.% WC ceramics is smaller.(2)In terms of mechanical properties, HfC-30 vol.% SiC with 5 vol.% WC shows higher performance, and deoxygenation of WC makes the sample have higher density; the increase in density and the formation of (Hf,W)C solid solution and the refinement of microstructure also make the hardness and flexural strength increase. The formation of (Hf,W)C solid solution leads to the increase in intergranular fractures, crack deflection, and crack bridging. Crack branching increases the energy dissipation during crack propagation, improving toughness.

## Figures and Tables

**Figure 1 materials-16-03337-f001:**
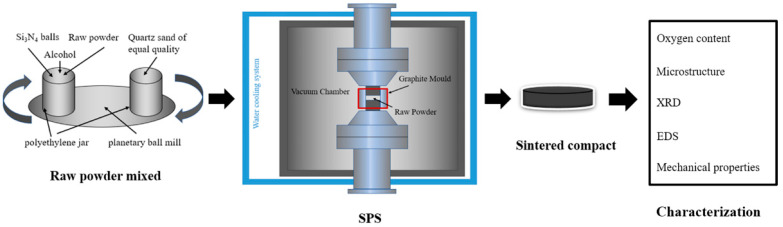
Graphical illustration of the experiment.

**Figure 2 materials-16-03337-f002:**
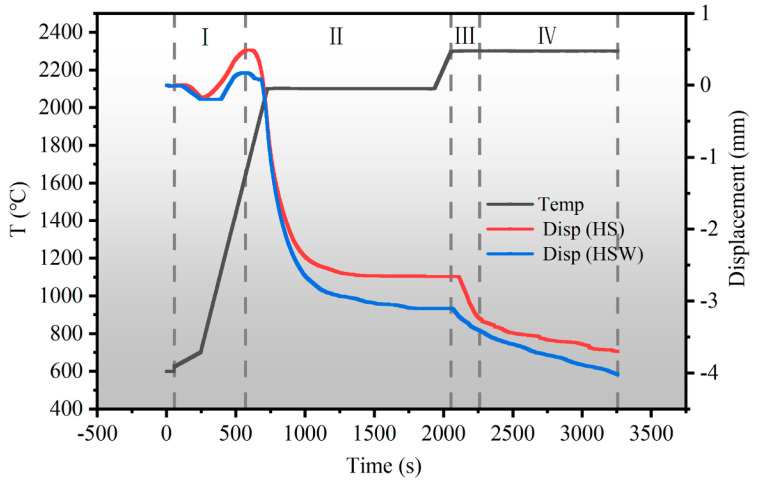
The displacement–temperature–time (DTT) diagram of HS and HSW. (I) First stage of sintering (II) Second stage of sintering (III) Third stage of sintering (IV) Fourth stage of sintering.

**Figure 3 materials-16-03337-f003:**
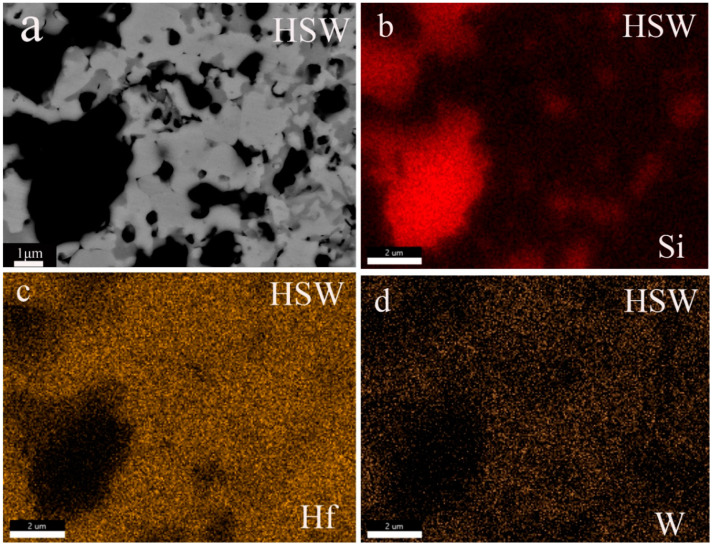
Surface image and corresponding Si, Hf, and W EDS maps. (**a**) SEM image at high magnification of the HSW (**b**) The maps of Si elements (**c**) The maps of Hf elements (**d**) The maps of w elements.

**Figure 4 materials-16-03337-f004:**
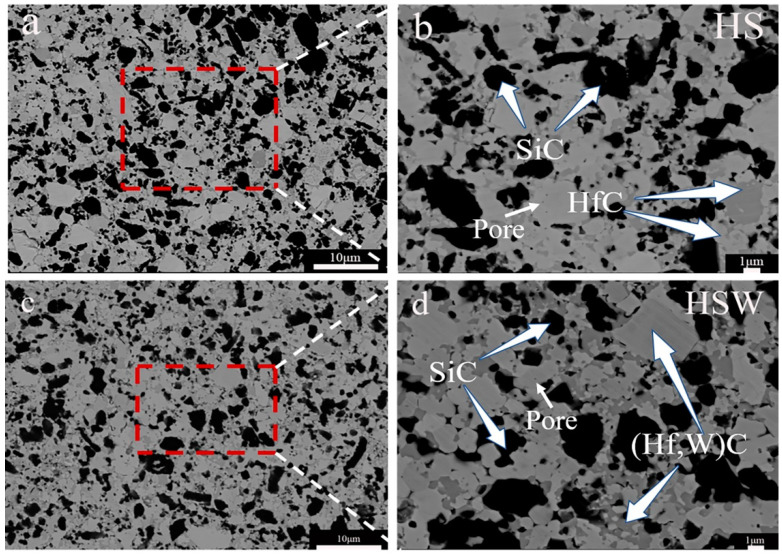
(**a**) SEM image of the HS composites microstructure; (**b**) SEM image at high magnification of the HS; (**c**) SEM image of the HSW composites microstructure; (**d**) SEM image at high magnification of the HSW.

**Figure 5 materials-16-03337-f005:**
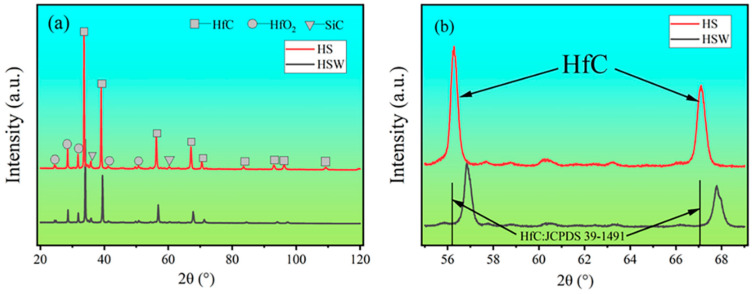
(**a**) XRD patterns of HS and HSW. (**b**) Amplification of diffraction peaks displacement of HS and HSW.

**Figure 6 materials-16-03337-f006:**
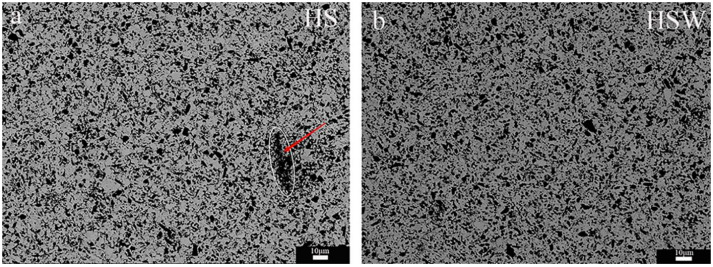
(**a**) SEM images of the microstructures of the HS in lower multiples. (**b**) SEM images of the microstructures of the HSW in lower multiples.

**Figure 7 materials-16-03337-f007:**
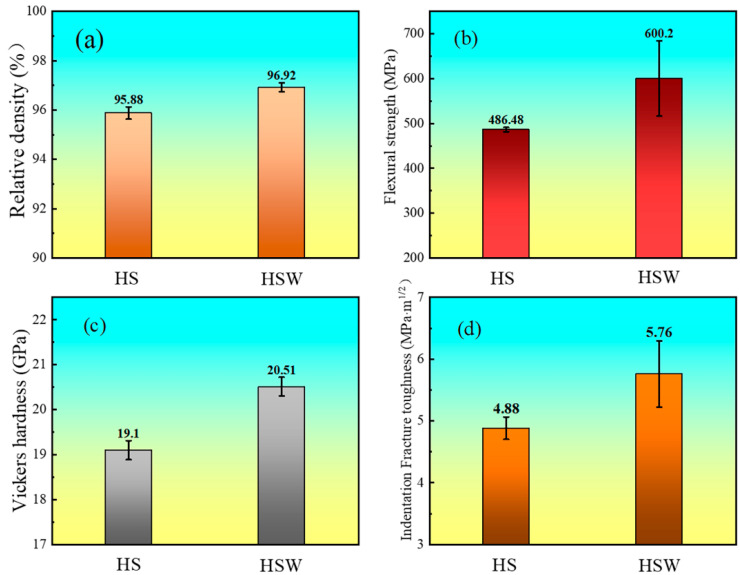
Relative density and mechanical properties of HS and HSW. (**a**) Relative density, (**b**) flexural strength, (**c**) Vickers hardness, and (**d**) indentation fracture toughness.

**Figure 8 materials-16-03337-f008:**
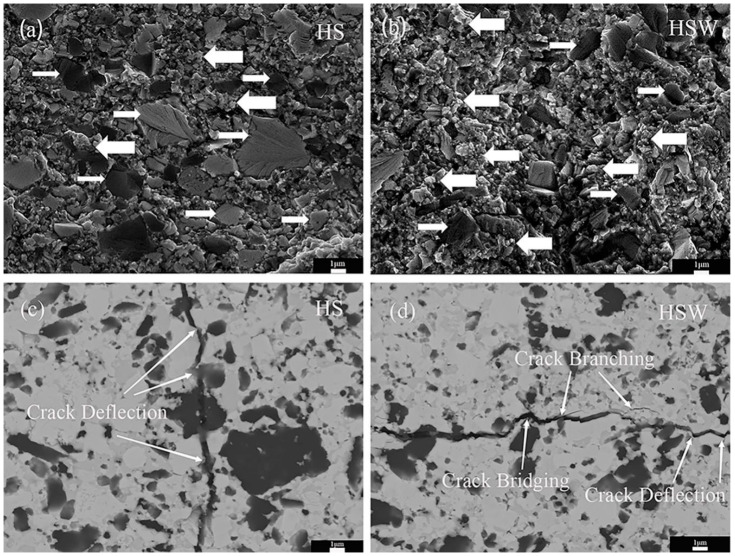
(**a**) SEM images of fractures of HS. (**b**) SEM images of fractures of HSW. (**c**) Indentation cracks of HS. (**d**) Indentation cracks of HSW. Thick arrows: intergranular fractures, narrow arrows: transgranular fractures.

**Table 1 materials-16-03337-t001:** Purity, major impurities, and average particle size of all raw powder.

Raw Powders	Purity (wt.%)	Major Impurities (wt.%)		Mean Particle Size (μm)
HfC	>99	Zr: 0.83		2
		Fe: 0.16		
SiC	>99	F. C: 0.42		0.5
		F. Si: 0.38		
WC	>99			0.2


## Data Availability

Data sharing is not applicable to this article.

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
