# Peer review of "Microstructure and Mechanical Properties of HfC-SiC Ceramics Influenced by WC Addition"

_materials, 2023, doi:10.3390/ma16093337_

Round 1
Reviewer 1 Report
The authors have investigated the effects of WC addition on microstructure and mechanical properties of HfC-SiC ceramics. The manuscript is generally in good condition. The methods are adequately described. The following comments are important to make it suitable for publication.
(1) please, add numerical and important results in abstract section.
(2) Schematic presentation can be added for materials and methods section.
(3) a visual research gap can be added after introduction section. It is believed that it will increase the impact of the paper.
(4) If it is possible, please add SEM image at high magnification of Figures 2a and 2b. In addition, it will be better to compare ratio of pores for two different compositions (HS, HSW). Because, relative density increases from 95.8% to 96.92%.
(5) authors say;
“ …… Figure 4a and b, the microstructure of HSW is relatively fine, and the addition of WC refines the microstructure of HSW …”
This sentence needs to be supported by references from literature if there is a similar study about WC refining the microstructure.
(6) The white arrows are defined in In Figures 6a and 6b. However, it is required to explain about arrows in the text.
(7) It is required to expand discussions and supporting results by references from literature.
(8) how are the relative density, hardness and mechanical results if compared to similar studies? It is important to show the highlights and advantages of the present study.
Reviewer 2 Report
The paper "Microstructure and mechanical properties of HfC-SiC ceramics influenced by WC addition" presents a relevant theme and within the scope of this journal, and can be considered after some corrections suggested below:
(a) The abstract is generally well written, however in terms of content it is generic, i.e., the authors lack an in-depth study of the quantitative results of this research;
(b) Scientific innovation is limited in the introduction of the paper, the authors must go deeper and detail what this research differs from countless others that exist on this topic, this must be evidenced together with the objectives at the end of the introduction;
(c) The state of the art of the evaluated topic needs to be improved by the authors, note that some topics are absent and need to be known with current research, such as: 10.1016/j.cscm.2020.e00437; 10.1016/j.cscm.2021.e00805;
(d) “Figure 5 shows the relative density and mechanical properties of HS and HSW. In Figure 5a, the addition of WC increases the relative density of the sample from (95.8 ± 0.2) % to (96.92 ± 0.18) %. The increase in relative density is attributed to the addition of WC, WC can reduce the oxide content on the particle surface and inhibit the growth of grains during sintering. Other researchers have reported the positive effects of the addition of MC (M = Zr, Ti, V...) on the sintering properties of MB2 composites, including the reduction of apparent porosity and the promotion of plastic deformation on surface and neck formation3” The authors should explain this passage more clearly.
(e) “Therefore, the toughening mechanism can be confirmed by crack propagation and fracture mode. Crack deflection, crack bridging and crack branching all increase the energy dissipation during the fracture process and thus increase the fracture toughness of materials” The authors should explain this passage more clearly.
(f) The form of citation presented is in disagreement with the general norms of the MDPI, the authors must review the entire document;
(g) The conclusion is not objective enough and does not show some of the main advances of this research.
Reviewer 3 Report
Reviewer Comment
Manuscript Number: Materials-2294860
Dear Editor,
The submitted manuscript is well prepared, successfully described the effect of addition of WC into HfC-30 vol. % SiC ceramics, and introduction is well arranged to shed a light how the starting research was conducted and why SPS is used based on conventional sintering technique.
1-The Abstract section should be rewritten and reflect the submitted article ore accurately
2-Introduction section: this correction is necessary:
as the raw POWDER and WC as the sintering aid
3- vol. % is the correct form and it should replace vol% throughout this submitted article.
4-Spark Plasma Sintering usually abbreviated as SPS, this should be used throughout this submitted article.
5-2.2 sintering: it can be replaced with
...and the heating rate was kept at 200 °C/min.
6- How authors judge the following statement?
When the sintering neck appears, a large number of holes and defects will be formed in the sintered body, which will affect the densification seriously.
Did the authors use SEM to check this stage of sintering?
7- Did the authors mean they used average data among five different readings for each property?
Five test rods were tested in each experiment.
8-for the section 2.3, sample characterisation,
for fracture toughness, equation (2) is the equation for the fracture toughness when Indentation Fracture (IF) technique is used. Currently in academia, IF technique accuracy is questionable nowadays. Even if consider for this fact that using this technique by the authors due to the difficulty using more advanced techniques such as SEVNB,...etc, the authors need to use the term of Indentation Fracture resistance (KIFR) Instead of fracture toughness KIC.
This fact, for example, was explained in detail in the following articles.
https://doi.org/10.1016/j.jeurceramsoc.2017.07.027
https://doi.org/10.1016/j.ijrmhm.2018.03.006
9-HV instead of Hv throughout the submited article.
10-for hot press, (HP) is a common known acronym. HT is not accurate.
11-for pressureless sintering, or conventional sintering, PLS is not an known acronym!
12-Does H.T. in section 3.1 refere to high temperature? if does, that is not a correct abbreviation.
13-section 3.2, DDT or DTT?
14-as the samples have relative density just above 95%, porosity should exist. However, the pores neither identified in the Figure 2 SEM micrographs and not explained in the section 3.3.
15-for EDS analysis, the percentage of each phase should be described.
16-The authors did not explain the effect of porosity (or decrease in porosity) in hardness.
17-in the reference section, chemical abbreviations should be used correctly. Just for example, ZrB2 should be corrected to ZrB2, La2O3 should be replaced with La2O3, and many other similar modifications are needed in the reference styles.
Reviewer 4 Report
In 3.3, page 6. Based on what results was it concluded that the SiC of HS is stable at 2300 ℃?
Round 2
Reviewer 3 Report
Reviewer Comment
Manuscript Number: Materials-2294860
Dear Editor,
The modifications made by authors make this submitted article worth to be published in your esteem journal.
Author Response
Response to Reviewer 3 Comments
Response to the comments:
The modifications made by authors make this submitted article worth to be published in your esteem journal.
Answer: Thank you for your reasonable, scientific and patient suggestions for revision, which will benefit me a lot from more rigorous scientific research in the future. In the process of revising the manuscript, your comments have been of a great help to me. Thank you again for your efforts on this manuscript.